# Cite Space-Based Bibliometric Analysis of Green Marketing

Li Liu 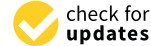, Hailang Cui and Yuankun Nie *

Business School, Yunnan University of Finance and Economics, Kunming 650221, China;
202102110154@stu.ynufe.edu.cn (L.L.); zz1771@ynufe.edu.cn (H.C.)
* Correspondence: zz0734@ynufe.edu.cn

**Abstract:** "Lucid waters and lush mountains are invaluable assets". Green marketing and sustainable development have become crucial topics in economic and social development. During 2019–2023, there are nearly 952 publications on green marketing-related topics in the Web of Science core database, and a large number of scholars researched green marketing, but there is still a lack of comprehensive and systematic studies on the current status of green marketing research and hot trends. The main purpose of this study is to summarize and sort out the current status of green marketing research by reviewing the literature related to green marketing (enterprises vs. consumers). Meanwhile, using Cite Space 6.2 R2 software, the core literature of green marketing in the past five years was screened, the data were visualized and analyzed, and a knowledge map of the cooperative authors, institutions, countries, and keywords was drawn in an attempt to discover the research hotspots and major development trends of green marketing. The results show that the publications and citation frequency of green marketing-related literature showed an increasing trend from 2019 to 2023, and the highest citation frequency was reached in 2022. Green marketing literature appears more frequently in the USA, China, and England and has a higher influence. Green management, firm performance, green innovation, green strategy, green capability, sustainable development, green business, green market orientation, green supply chain management, green exploitation, green responsibility, ecosystem, green commitment, green perceptions, green competitive advantage, and corporate social responsibility reflect the hot topics and important themes of green marketing research. This study innovatively combines a green marketing literature review and bibliometric analysis, comprehensively and systematically discusses the current status of green marketing research and hot trends based on both textual and data perspectives, and puts forward the "individual-environment-economy-society" virtuous ecological cycle of green marketing from a macro perspective.

**Keywords:** green marketing; bibliometric analysis; virtuous ecological cycle of "individual-environment-economy-society"

## 1. Introduction

"Lucid waters and lush mountains are invaluable assets" is an important assertion made by Chinese President Xi Jinping that aligns with the significant issue of green marketing and sustainable development. In 2022, the resumed Fifth United Nations Environment Assembly (UNEA5.2) called on the international community to unite for a peaceful and healthy environment, focusing on global economic recovery and sustainable development and emphasizing the importance of strengthening nature's actions to achieve Sustainable Development Goals, which are key to personal, economic, environmental, and social sustainability. Focusing on the marketing perspective, green marketing and sustainability have received a great deal of attention from scholars and are important hot trends in green marketing research in the new era.

Green marketing includes ecological marketing, environmental marketing, marketing of resource-saving and environmentally friendly products [1], and marketing that meets the needs of consumers and enterprises in the present while preserving or enhancing

inter-generational legacy [2]. Green marketing integrates personal, economic, and social considerations and is an important tool and marketing approach to promote resource conservation, environmental protection, and ecological development. From the enterprise perspective, green marketing focuses on corporate social responsibility [3], sustainability marketing [4], socially responsible marketing [5], green marketing orientation [6], green marketing [7], corporate green product innovation [8], green marketing strategies [9], and so on. The research topics are very rich, but there is a lack of influence of other social subjects, such as government, consumers, the public, and corporate employees, on green marketing. There is also no research to think deeply about the possible negative effects of external environmental pressure and the market dynamics of green marketing. From the perspective of consumers, green marketing is mainly manifested in consumers' awareness and behaviors of practicing green consumption, and some scholars point out that consumers' pro-environmental behaviors [10,11] and consumers' innovativeness [12,13] influence consumers' willingness to green consumption. However, few studies have explored the deviation of green consumption behaviors from consumers' actual purchases, as well as consumers' willingness to repeat green consumption, and there is a lack of research on consumers' repeated sustainable consumption of green products.

Based on the importance of green marketing research and the current status of existing research, scholars have gradually developed quantitative research methods for literature reviews by shifting from qualitative analysis of literature reviews to econometric analysis of the literature. Initially, most scholars conducted the review analysis of green marketing through literature combing [14]. With the research development of the bibliometric analysis method, the quality and quantity of the literature were visualized by analyzing the authors, publishing institutions, countries, and citation frequency [15]. Bibliometric analysis intuitively and concisely reflects existing research trends and hot topics, but few scholars use data visualization analysis for green marketing research [16], and few studies consider green marketing from a macro perspective of sustainability.

Therefore, this study aims to use the data visualization software Cite Space, using the SCI and SSCI source journals of the Web of Science core database as the main data source, and "Green marketing" or "Environmental Marketing" or "Lower-Carbon Marketing" or "Ecological Marketing " or "Sustainable Development" as a broad keyword search to conduct a comprehensive and systematic bibliometric study of green marketing. Through the mapping of co-authors, institutions, countries, and keywords, this study innovatively combines a green marketing literature review and literature data analysis to discuss the current status of green marketing research and hot trends comprehensively and systematically based on both textual and data perspectives, providing practical references for subsequent scholars' research selection and green marketing theory development. At the same time, the study will also specifically explore the following issues:

(1) What is the current status of research on green marketing research?
(2) What is the overall trend of green marketing research from 2019 to 2023?
(3) Which authors, publishers, and countries have a higher influence in the field of green marketing?
(4) What are the hot keywords in green marketing?
(5) What are the future trends in green marketing research?

## 2. Literature Review

### 2.1. Green Marketing

In 1969, Lazer first introduced the concept of green marketing into the social dimension of marketing, taking into account the limited resources and the need to transform traditional marketing methods [17]. Green marketing is an emerging marketing approach that meets the current needs of consumers and enterprises while preserving or enhancing inter-generational inheritance capabilities [1,2]. It is rooted in a comprehensive consideration of individuals, society, and nature, and it is an important means and form of promoting ecological development and environmental protection. Consumers and enterprises are the

main targets and agents of green marketing, and the existing literature has studied the subject from both perspectives [14] (Table 1).

**Table 1.** Conceptualisation of green marketing in this paper.

| Angle of View | Main |
| --- | --- |
| Consumer perspective | Based on the cognitive-affective-behavioral theory, consumers' green marketing behaviors can be divided into three levels: at the cognitive level, consumers' green marketing behaviors are influenced by changes in their perceptions of green marketing concepts, motivations for transmission, and attention to future sustainable development; at the affective level, it includes consumers' green emotions, green perceived value, and other emotional elements; at the behavioral level, it is manifested in consumers' green consumption preferences, green purchase intentions, attitude-purchase behavior deviations, and other behavioral intentions. |
| Corporate perspective | While satisfying their economic interests, enterprises also consider their social responsibility towards nature and ecology, creating a dual benefit of economic and social benefits to achieving sustainable development strategies. |

*2.2. Green Marketing Subjects: Enterprises*

Green marketing and sustainable development are market-driven [4]. Most scholars mainly focus on formulating and implementing green marketing from the perspectives of both the enterprise and the consumer in the buyer–seller relationship.

Green marketing orientation is a strategic awareness at the ideological level, which is an important environmental strategic orientation of enterprises, covering subjective cognition, such as corporate social responsibility and internal employee green values, in terms of values [18], thus achieving sustainable development of corporate development goals. Enterprises with a green marketing orientation can bring economic performance themselves [19]. For example, some scholars have pointed out that green marketing orientation can realize the green marketing management of hotels, and consumers' sustainable consumption behaviors affect society's sustainable development. The green marketing orientation model established by green hotels is not only a comprehensive expression of corporate social responsibility but also an important means to improve consumers' reputation and loyalty to hotels [20]. Due to the limitations of traditional technology, market-oriented green innovation models have been widely used by scholars [21]. Green innovation actively promotes green sustainable development and serves as an important predictive indicator of sustainable development [22]. The existing literature on the green marketing orientation of enterprises mainly focuses on the importance of green marketing orientation and its connection with sustainable development but lacks an in-depth consideration of the impact mechanism of other social entities, such as government, consumers, the public, and enterprise employees, on the green marketing orientation of enterprises.

Green marketing capability is the enterprises' ability to identify and implement green marketing [23], and enterprise identification is to identify external environmental pressures, including resource scarcity, policy guidance, competitive pressure, and market changes; enterprise implementation is the practice of marketing capabilities through the integration of internal and external resources, including product, price, channel, promotion, and other marketing strategies [24]. The existing literature points out that by sustainability-oriented dynamic capabilities, enterprises apply the comprehensive effects of external resource integration, internal and external resource identification integration, and resource construction and reallocation to promote green product innovation and green product marketing and obtain good market feedback for green products [25]. The sustainability-oriented eco-innovation capability helps enterprises to meet the rapidly changing regulatory, technological, and market demands [26]. Enterprises also include the ability to voluntarily implement environmental management policies, voluntarily undertake social responsibili-

ties, ecological research, and development, innovation capabilities, green market perception capabilities [27], and green marketing practice capabilities. The realization of enterprise green marketing capabilities is dedicated to meeting consumers' green consumption needs. Existing research on the green marketing capabilities of enterprises exists from various perspectives, but the core is centered on green and sustainability as the fundamental goals.

The green marketing effect is an important implementation manifestation of enterprise green marketing orientation and green marketing capability, which is reflected in green product marketing and social benefits. Green marketing promotes socially sustainable development, and research on public sectors and manufacturing considers that green marketing has significant positive effects and can bring benefits to resource conservation and environmental protection [28,29]. Some scholars have also pointed out that government policies, government tax incentives, and enterprise green production investments significantly affect the welfare of enterprises, markets, and society [30]. Meanwhile, considering the enterprises' reputation as an important intangible asset of marketing performance and financial performance, existing research has found a close relationship between environmental performance, financial performance, and corporate reputation [31,32]. Therefore, green marketing not only promotes the enterprises' sustainable development and creates economic benefits for enterprises but also helps enterprises to maintain a good social image, create social benefits, and achieve the dual performance of economy and society.

Green marketing orientation, green marketing capability, and green marketing effect mainly reflect the enterprises' strategic thinking, green capabilities, and green marketing implementation. The existing literature on the enterprise perspective of green marketing mainly reflects two aspects: on the one hand, many scholars emphasize the awareness of corporate green marketing, sustainable marketing, and social responsibility marketing, and, on the other hand, existing research emphasizes the positive effects of the corporate implementation of green marketing awareness, reflected in financial performance, corporate reputation, and consumer support.

### 2.3. Green Marketing Subjects: Consumers

During the process of making green product purchasing decisions, consumers are influenced by factors such as green satisfaction, green trust, green reputation, and green perceived value, resulting in varying degrees of green purchase intentions [33]. Consumer experience value, customer relationship quality, and customer loyalty also have positive effects on green consumption [34]. These studies not only reflect the effectiveness of enterprise green marketing but also the emotional attitudes and behavioral intentions of consumers towards enterprise green products. Based on themselves, consumers with green consumerism characteristics are more willing to purchase environmentally friendly products or products with ecological labels, thus making green purchasing decisions [35]. Some scholars have also pointed out that consumers' purchasing behaviors may be motivated by altruism (vs. egoism), and altruism often manifests as pro-environmental behavior [36], and consumers are willing to purchase green products based on this motivation. Existing research mainly focuses on the influence of enterprise green product characteristics on consumer green cognition, emotions, and behavior; however, there is a lack of research on the mechanism of consumer attitudes-behavior intentions and behavior intention-actual behavior deviations towards green consumption, and there are also few studies on the impact of consumers on enterprise green marketing performance.

## 3. Methodology

### 3.1. Method

The Cite Space data visualization software developed by the scholar Chen comprehensively integrates social network analysis, cluster analysis, and other analytical methods. The software's visualization analysis of authors, cited references, keywords, time slices, and other factors can effectively explain the research frontiers and emerging trends in the field [37,38]. This study used bibliometric methods, focusing on the core foreign literature related to green

marketing. From a macro perspective, this research aimed to understand the current status and development trends of green marketing and sustainable development.

### 3.2. Data Source

The Web of Science database contains vast literature data resources and is highly recognized and selected by scholars. Therefore, this study mainly used the Web of Science Core Collection database, including SCI and SSCI source journals (Table 2). First, the WOS database is considered the most reliable and resource-rich database. Second, the SCI and SSCI source journals are of high quality and can effectively reflect the current status and development trends of green marketing research. The authors searched for the keywords "Green marketing" in the Green marketing-related subject terms in the WOS Core Collection database, with the search term input as "TS = ("Green marketing" or "Environmental Marketing" or "Lower-Carbon Marketing" or "Ecological Marketing" or "Sustainable Development") (Table 2).

**Table 2.** Literature retrieval rules.

| Key | Details |
| --- | --- |
| Retrieval Time | 3 June 2023 |
| Data Source | Web of Science core database—SCI, SSCI |
| Retrieve Topic | "Green Marketing" or "Environmental Marketing" or "Lower- Carbon Marketing" or "Ecological Marketing" or "Sustainable Development" |
| Time Zone | Last 5 years (January 2019–June 2023) |
| Screening | 1. Research direction is limited and related to green marketing<br>2. Publishing journal related to marketing<br>3. Cite Space weight removal function |
| Search Results | 952 valid references, including 9723 citations |

## 4. Data Analysis

### 4.1. Overall Trends Analysis

The authors searched the green marketing-related literature published from 2019 to 2023 and found an increasing trend in publication and citation frequency. In 2021, the number of publications related to green marketing increased and reached a peak citation frequency in 2022 (Figure 1). These findings indicate that green marketing has been a significant area of interest and focus for scholars in recent years.

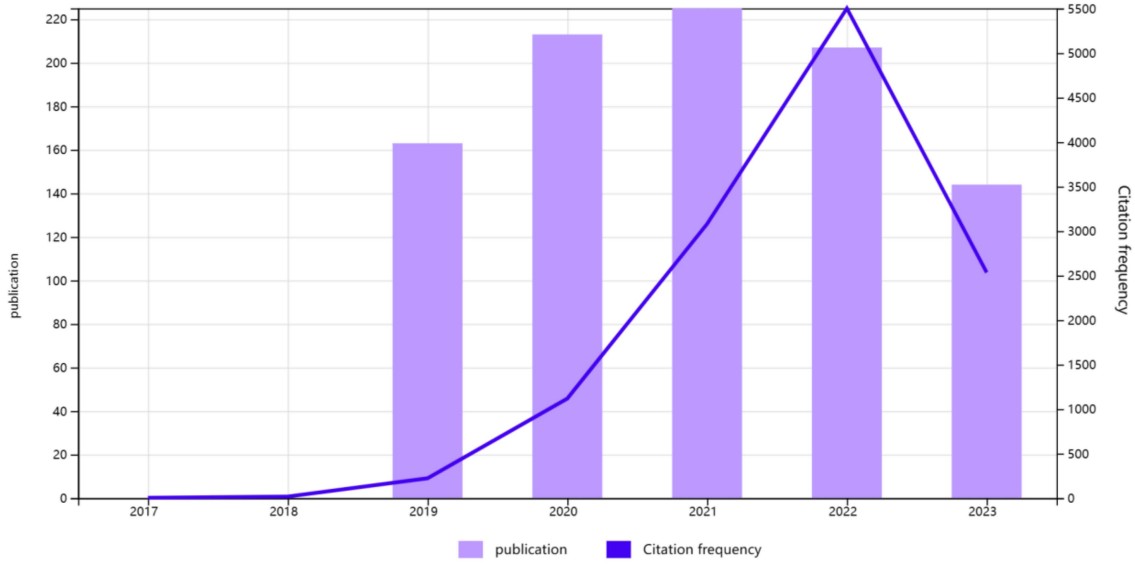

**Figure 1.** Green marketing research publications and citation frequency.

## 4.2. Research Subjects Analysis

By analyzing the co-occurrence of author, publishing institution, and country data, the results show that the countries with high count rates in the field of green marketing include the USA, China, England, Spain, Australia, Italy, France, India, Germany, Sweden, Finland, Canada, Brazil, Taiwan (China), and the Netherlands (Table 3). Among them, the countries with the highest impact are mainly the USA, China, and England. In terms of institutions, the Indian Institute of Management, the White Rose University Consortium, the University System of Ohio, the University System of Florida, Xi'an Jiaotong University, and other important organizations have a high citation rate for green marketing literature, indicating important sources of authors in green marketing research.

**Table 3.** Top fifteen influential countries and publishing institutions in green marketing research.

| Number | Countries | Count | Year | Institutions | Count | Year |
|---|---|---|---|---|---|---|
| 1 | USA | 207 | 2019 | Indian Institute of Management | 17 | 2019 |
| 2 | CHINA | 195 | 2019 | White Rose University Consortium | 15 | 2020 |
| 3 | ENGLAND | 139 | 2019 | University System of Ohio | 15 | 2019 |
| 4 | SPAIN | 84 | 2019 | University System of Florida | 13 | 2019 |
| 5 | AUSTRALIA | 69 | 2019 | Xi'an Jiaotong University | 12 | 2019 |
| 6 | ITALY | 69 | 2019 | Aarhus University | 9 | 2019 |
| 7 | FRANCE | 66 | 2019 | Chinese Academy of Sciences | 8 | 2020 |
| 8 | INDIA | 51 | 2019 | Arizona State University | 8 | 2019 |
| 9 | GERMANY | 44 | 2019 | University of Newcastle | 7 | 2022 |
| 10 | SWEDEN | 41 | 2019 | University System of Georgia | 7 | 2020 |
| 11 | FINLAND | 41 | 2019 | South China University of Technology | 7 | 2020 |
| 12 | CANADA | 38 | 2019 | Loughborough University | 7 | 2021 |
| 13 | BRAZIL | 36 | 2019 | Kristiania University College | 7 | 2019 |
| 14 | TAIWAN, CHINA | 32 | 2019 | Indian Institute of Technology System (IIT System) | 7 | 2022 |
| 15 | THE NETHERLANDS | 30 | 2019 | Arizona State University-Tempe | 7 | 2019 |

## 4.3. Research Hotspots Analysis

Keyword co-occurrence is an intuitive way to illustrate the hotspots and temporal context of related research, which is highly condensed in the existing literature. The keyword co-occurrence function in the Cite Space software can identify keywords and recognize emerging trends and development patterns in existing research [37,39]. The g-index refers to the cumulative times that a paper is cited among papers with at least g citations, indicating the maximum serial number of the most frequently cited papers. The cumulative citation of the $(g + 1)$ paper will be less than $(g + 1)^2$, $g^2 \leq k\Sigma ci$, $k \leq (g + 1)^2$, generally with the scale factor k = 25 [39].

To further understand the research directions of green marketing and sustainable development, this study used keyword co-occurrence analysis to reveal the hotspots and underlying connections. In the Cite Space software, the nodes were set as keywords, the g-index method was selected, and k was set to 25. After visualizing the results, a knowledge map of high-frequency keywords related to green marketing and sustainable development was generated. The map contained 374 nodes, 2405 links, and a network density of 0.0345 (N = 374, E = 2405, D = 0.0345) (Table 4). Each circular node represents a keyword, and the size of the node represents the frequency of keyword co-occurrence. The overlapping areas between the nodes represent high-frequency clusters of keywords. The network density indicates the modularity of the network, and a higher value indicates better clustering effects, further verifying the rationality of keyword clustering analysis. Similarly, Modularity (Q) > 0.3 and Silhouette (S) > 0.5 also indicated better clustering [27], which again verified the rationality of keyword clustering analysis. Therefore, the nine high-frequency keywords that appeared most frequently included corporate social responsibility, entrepreneurial orientation, environmental innovation, innovation governance, sustainable supply chain management, relational governance, supplier code, firm innovation, and

sustainable development goal (Figure 2). These keywords reflected the main research hotspots of green marketing.

**Table 4.** Data examination.

| Time Span | N | E | D | Q | S | (Q, S) |
|---|---|---|---|---|---|---|
| 2019–2023 (Slice Length = 1) | 374 | 2405 | 0.0345 | 0.3766 | 0.693 | 0.488 |

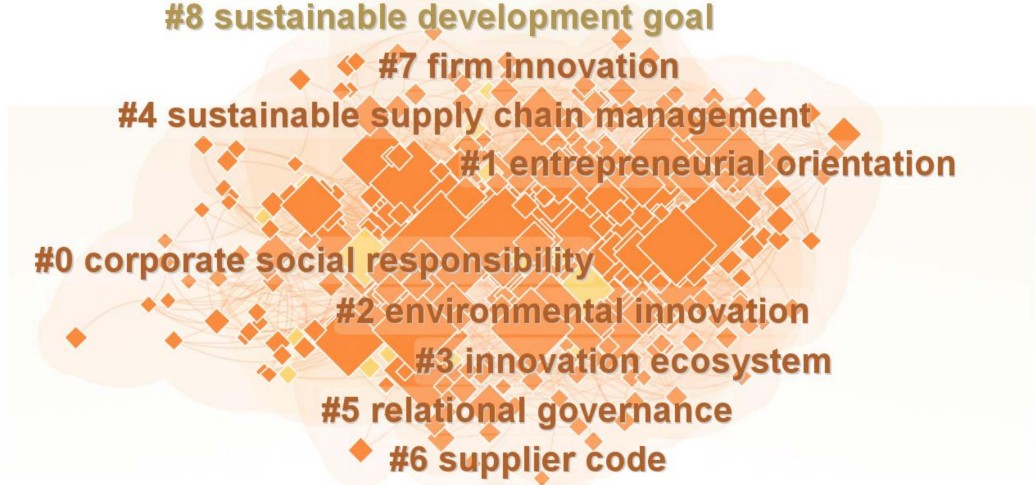

**Figure 2.** Knowledge map of keywords.

Further analysis of the frequency of keyword appearances revealed that there were 122 keywords with a frequency of more than 10 in the field of green marketing research, and the top 15 high-frequency keywords included green management, firm performance, green innovation, green strategy, green capability, sustainable development, green business, green market orientation, and green supply chain management (Table 5). These high-frequency keywords all reflected the research hotspots of green marketing. In addition to co-occurrence frequency, centrality is an important indicator of a node's connection ability with other nodes [37]. Centrality generally ranges from 0 to 1, and higher centrality can reveal important topics in existing research to a greater extent [40]. Therefore, based on the centrality ranking, 171 keywords had a high centrality (C > 0.01). The high-centrality keywords, such as green exploitation, green responsibility, ecosystem, green commitment, green perceptions, and corporate social responsibility, indicated the core research themes of related studies.

**Table 5.** Green marketing research keywords.

| Number | Keywords | Count | Year | Keywords | Centrality | Year |
|---|---|---|---|---|---|---|
| 1 | Green management | 168 | 2019 | Green exploitation | 0.05 | 2019 |
| 2 | Firm performance | 144 | 2019 | Institutional theory | 0.05 | 2019 |
| 3 | Green innovation | 123 | 2019 | Green conceptual framework | 0.05 | 2019 |
| 4 | Green strategy | 120 | 2019 | Business models | 0.05 | 2019 |
| 5 | moderating role | 91 | 2019 | Green responsibility | 0.04 | 2019 |
| 6 | financial performance | 90 | 2019 | Green business performance | 0.04 | 2019 |
| 7 | Green capability | 86 | 2019 | Green organizations | 0.04 | 2019 |

**Table 5.** *Cont.*

| Number | Keywords | Count | Year | Keywords | Centrality | Year |
|---|---|---|---|---|---|---|
| 8 | Sustainable development | 84 | 2019 | Green commitment | 0.04 | 2019 |
| 9 | dynamic capability | 81 | 2019 | Green perceptions | 0.04 | 2019 |
| 10 | Sustainability | 81 | 2019 | organization | 0.04 | 2019 |
| 11 | Green business | 80 | 2019 | Green market orientation | 0.03 | 2019 |
| 12 | resource Resource-based view | 74 | 2019 | Green supply chain management | 0.03 | 2019 |
| 13 | Green market orientation | 73 | 2019 | mediating role | 0.03 | 2019 |
| 14 | Green supply chain management | 73 | 2019 | Green competitive advantage | 0.03 | 2019 |
| 15 | Enterprise framework | 72 | 2019 | Corporate social responsibility | 0.03 | 2019 |

Through data analysis of keyword co-occurrence and centrality, this study found that green market orientation, corporate social responsibility, green supply chain management, green business performance, and green perceptions were both high-frequency and high-centrality keywords, reflecting the research hotspots and major themes of existing research.

*4.4. Citation Bursts of Green Marketing Literature*

By using the burst detection feature of the Cite Space software, sudden increases or decreases in the number of citations for specific keywords can be detected [41]. The blue period indicates the last five years of data collection, and the red period indicates the period when the articles were highly cited [42,43]. The data results show that green strategic alliances, green knowledge transfer, green implementation, and green uncertainty have higher impact intensity in the literature related to corporate green marketing research. Sustainable innovation, green scale development, and green growth have a longer duration of impact (Table 6).

**Table 6.** Top 25 keywords with the strongest citation bursts.

| Number | Keywords | Year | Strength | Begin | End | 2019–2023 |
|---|---|---|---|---|---|---|
| 1 | Green strategic alliances | 2019 | 2.87 | 2019 | 2020 | |
| 2 | Green knowledge transfer | 2019 | 2.58 | 2019 | 2020 | |
| 3 | Green implementation | 2019 | 2.51 | 2019 | 2020 | |
| 4 | Dominant logic | 2019 | 2.29 | 2019 | 2020 | |
| 5 | Relationship quality | 2019 | 2.01 | 2019 | 2020 | |
| 6 | Green uncertainty | 2019 | 2.01 | 2019 | 2020 | |
| 7 | Green marketing strategy | 2019 | 1.72 | 2019 | 2020 | |
| 8 | Identification | 2019 | 1.72 | 2019 | 2020 | |
| 9 | Slack resources | 2019 | 1.72 | 2019 | 2020 | |
| 10 | Green adoption | 2019 | 1.64 | 2019 | 2020 | |
| 11 | Contingency theory | 2019 | 1.43 | 2019 | 2020 | |
| 12 | Green alliances | 2019 | 1.43 | 2019 | 2020 | |
| 13 | Human resource management | 2020 | 2.01 | 2020 | 2021 | |
| 14 | Green business model | 2020 | 2.01 | 2020 | 2021 | |

**Table 6.** *Cont.*

| Number | Keywords | Year | Strength | Begin | End | 2019–2023 |
|---|---|---|---|---|---|---|
| 15 | Power | 2020 | 1.56 | 2020 | 2021 | |
| 16 | Radical innovation | 2020 | 1.56 | 2020 | 2021 | |
| 17 | Green opportunity | 2020 | 1.34 | 2020 | 2021 | |
| 18 | Eco-innovation | 2021 | 3.79 | 2021 | 2023 | |
| 19 | Sustainable innovation | 2021 | 2 | 2021 | 2023 | |
| 20 | Green scale development | 2021 | 1.78 | 2021 | 2023 | |
| 21 | Green ownership | 2021 | 1.56 | 2021 | 2023 | |
| 22 | Green growth | 2021 | 1.33 | 2021 | 2023 | |
| 23 | Business relationships | 2021 | 1.33 | 2021 | 2023 | |
| 24 | Manufacturing firms | 2021 | 1.33 | 2021 | 2023 | |
| 25 | Green markets | 2021 | 1.33 | 2021 | 2023 | |

Notes: The bolded time points from 2019 to 2023 indicate the temporal intensity of the paper's ongoing impact.

## 5. General Discussion

### 5.1. Green Marketing Overall Trends

Green marketing and sustainable development are an important topic for economic and social development. From the data of bibliometrics, there were 952 articles and 9723 citation data of related literature about green marketing in 2019–2023, and green marketing was paid attention to by a large number of scholars. Similarly, the literature related to green marketing shows the development trend of increasing year by year, and the academic community still maintains high attention to the research on green marketing. From the perspective of authors, institutions, and countries, the USA, China, and England, as countries with high influence in the field of green marketing research, can be cited as important sources of green marketing literature. The Indian Institute of Management, the White Rose University Consortium, the University System of Ohio, the University System of Florida, Xi'an Jiaotong University, etc. include important authors with high citation rates in green marketing literature; therefore, the green marketing literature of the above-mentioned institutions can be a good source for other green marketing researchers. The green marketing literature of these institutions can be a good inspiration and reflection for other green marketing researchers.

### 5.2. Green Marketing Keywords

Upon reviewing the analysis of the literature review, the authors found that scholars mostly think about green marketing from enterprise (vs. consumer) perspectives [4].

The enterprise perspective of green marketing focuses on green marketing orientation [18–22], green marketing capability [23–27], and green marketing effectiveness [28–32], which mainly include corporate green marketing awareness and green marketing formulation and implementation. Furthermore, through bibliometrics analysis, it was found that the keywords with a high co-occurrence include corporate social responsibility, entrepreneurial orientation, environmental innovation, innovation governance, sustainable supply chain management, relational governance, supplier code, firm innovation, and sustainable development goal, which are in line with the enterprise perspective of current green marketing research. Furthermore, the high-frequency keywords of green marketing research include green management, firm performance, green innovation, green strategy, financial performance, green capability, sustainable development, green business, green market orientation, and green supply chain management. The high-frequency keywords for green marketing research include green conceptual framework, green responsibility, green business performance, green organizations, green commitment, green perceptions, green market orientation, green supply chain management, green competitive advantage, and corporate social responsibility. Therefore, the high-frequency keywords (vs. high-centrality keywords) that overlap with keywords such as green market orientation, corporate social responsibility, green supply chain management, and green business performance can be

the key themes of green marketing focus. However, the existing studies of green marketing from an enterprise perspective have not deeply considered the possible external environmental pressures and the negative effects of the market dynamics of green marketing, but the data results show the high impact intensity of green uncertainty and slack resources keywords. This indicates that the uncertainty of resource scarcity and green development as the external environmental pressures of green marketing has an important role, which fills an important gap in green marketing research.

The consumer perspective on green marketing focuses on the possible green consumption decisions and purchase intentions of consumers [33–36]. The overlapping keywords green perceptions with high-frequency keywords (vs. high-centrality keywords) also reflect the importance of green consumers in green marketing research, but the literature on green consumers is cited and co-occurrence is less frequent.

## 6. Conclusions

### 6.1. Theoretical Implications

Based on the literature review and measurements, this study finds that the main subjects of green marketing research include enterprises and consumers. The study aims to innovatively sort out the virtuous cycle of "individual (consumer vs. company)-environment-economy-society" green marketing and sustainability and provide a macroscopic understanding of the text and data of the literature studies.

From the perspective of consumers, green marketing is a new marketing model and marketing is practice oriented. The practice results of green marketing depend on the two-level attitudes accepted by consumers, which are influenced by consumer cognition, emotion, and behavior. Green moral motivation drives consumers' environmental moral identity and green personal norms [44], and consumers identify, understand, and actively generate pro-environmental behaviors, consciously constraining their green responsibility awareness and behavioral norms. For green products, consumers' perception quality and perception value directly affect their behavioral performance. Green perception quality refers to consumers' overall perception and judgment of the quality or other superiority of green products and services provided by companies [45,46], while green perception value refers to the gains and losses perceived by consumers when making purchasing decisions for green products [47,48]. Both perception quality and perception value significantly affect consumers' green consumption preferences and purchase intentions, thereby achieving a virtuous ecological cycle of "consumer-environment-economy-society".

From the perspective of enterprises, the green marketing performance and sustainable development configuration of enterprises are significantly affected by green marketing capability (GMC), which is the ability of enterprises to identify, acquire, convert, and integrate existing resources to create products and services that meet the environmental requirements of enterprises, with two key elements of green market perception and green marketing implementation [23,25]. Green market perception is the overall planning and strategic goal setting of enterprises for the green market and timely learning based on dynamic changes in the market, such as consumers and competitors [49]. From an overall strategic perspective, enterprises fully consider corporate social responsibility, sustainable strategic goals, and sustainable development. Green marketing implementation achieves the established goals of enterprises through market marketing strategies, including green product design and development, green pricing considering economic and environmental costs, green distribution channels for sustainable supply chain management, green promotion methods that convey information such as green and environmental protection of enterprise products to the outside world, as well as integrating cross-functional sustainable innovation and development capabilities within the enterprises [50,51]. Based on macro and micro measures, such as environmental recognition, strategic formulation, and marketing implementation, enterprises achieve a virtuous ecological cycle of "enterprise-environment-economy-society".

*6.2. Research Limitations*

This study also has certain research limitations. On the one hand, the research objects mainly include the SCI and SSCI source journals in the Web of Science core database in the past five years, with reliable data sources and convincing data analysis results [52,53]. However, the research data mainly included authors, institutions, countries, and keywords. Due to the time limit, the study failed to sort out the timeline of green marketing research and did not comb the development context of green marketing research [54,55]. On the other hand, the analysis of the existing literature using Cite Space software belongs to the objective analysis category and is not influenced by personal subjective experience. However, there may still be data errors in the literature research.

*6.3. Future Perspectives*

This study also provides new thinking for the future of green marketing. By analyzing the sudden increase or decrease in the number of keyword citations in a specific period, the study found that eco-innovation, sustainable innovation, green scale development, green ownership, green growth, business relationships, manufacturing firms, and green markets have a longer time duration of impact. Therefore, in addition to a green marketing orientation, green marketing capability, green marketing effectiveness, and green consumers, scholars can think about the impact of eco-innovation and sustainable innovation on individuals (enterprises vs. consumers), the economy, and society in future studies.

**Author Contributions:** Conceptualization, Y.N.; Methodology, L.L.; Formal analysis, H.C.; Data curation, H.C. All authors have read and agreed to the published version of the manuscript.

**Funding:** This research received no external funding.

**Institutional Review Board Statement:** Not applicable.

**Informed Consent Statement:** Not applicable.

**Data Availability Statement:** Not applicable.

**Conflicts of Interest:** The authors declare no conflict of interest.

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
