# Peer review of "Cite Space-Based Bibliometric Analysis of Green Marketing"

_sustainability, doi:10.3390/su15129840_

Round 1

Reviewer 1 Report (Previous Reviewer 1)

The author has made many revisions, and the current version of the manuscript shows a significant improvement in narrative compared to the previous version. My specific comments are as follows:

You quoted a statement by President Xi at the beginning of the article, but in Figure 2, Taiwan is listed as a separate country. It made me smile. I suggest that you indicate "Taiwan, China" in Figure 2 and modify the image name accordingly.

Why do some countries in Figure 2 show institutions while others do not? Why are these specific institutions displayed? If you want to reflect the publication output of institutions, I suggest creating a separate chart for that. The current version of the image makes it difficult to visually represent the publication output of countries and institutions clearly.

The text on the images is difficult to read. For example, the small text in the top left corner of Figure 2 and Figure 3 should be enlarged if you consider them essential information to be displayed. Additionally, the colors in Figure 3 are excessively strong. I recommend adjusting the visual settings to make the text easier to read, such as adding shadows or using other methods. The color saturation and brightness are too high, which strains my eyes.

Before using some specialized terms, you need to explain what they represent and how they are calculated. For example, g index, network density, centrality.

Why is c > 0.01 considered a high level? Please provide literature references to support this.

Why are the 12 high-frequency keywords in lines 100 to 104 different from the 15 keywords in lines 107 to 116? What is the difference between them? How should these two sets of terms be understood in theoretical exploration or practical applications? Which is more important?

Your discussion and conclusion sections are written like a literature review. This is incorrect, as they should not appear here, and some content can even be omitted from the manuscript. You need to rewrite the content of the discussion and conclusion sections. You should articulate your research contributions and value from both theoretical and practical perspectives. Additionally, you need to provide a more critical interpretation of your research results. I provide an example for your reference when rewriting these sections: [link to example paper] (https://doi.org/10.1371/journal.pone.0264562).

The literature review section is missing. I suggest that you conduct literature reviews and provide operational definitions for the important dimensions discovered. Include these contents in Chapter 2. For example:

2. Literature Review

2.1 xxx

2.2 xxx

The section on future research is a bit verbose. I suggest streamlining it. Focus on discussing your research limitations and future research directions.

Author Response

Reviewer 2 Report (Previous Reviewer 3)

The Authors have tried to revise the article and the manuscript has
been improved.

Author Response

Response to Reviewer 2 Comments

Point 1: The Authors have tried to revise the article and the manuscript has been improved.

Response 1:

Thank you very much for your suggestions. I will make further corrections to the logical order and the English language, and I very appreciate your valuable suggestions.

This is a reorganized abstract of the paper:

"Lucid waters and lush mountains are invaluable assets". Green marketing and sustainable development have become crucial topics in economic and social development. This study utilized the Cite Space software to visualize and analyze relevant literature published in the Web of Science core database from 2019 to 2023, drawing knowledge maps of collaborating authors, institutions, countries, and keywords to identify the research hotspots and development directions of green marketing. The research results show an increasing trend in the publication and citation frequency of green marketing-related literature from 2019 to 2023, with the highest citation frequency achieved in 2022. The relationship between green management, Firm performance, green innovation, green strategy, green capability, Sustainable development, green business, green market orientation, green supply chain management, green exploitation, green responsibility, Ecosystem, Green commitment, green perceptions, Green competitive advantage, Corporate social responsibility reflect important hotspots of green marketing research. From a microscopic perspective, the study sorts out concepts related to green marketing, discusses the virtuous ecological cycle of "individual-environment-economy-society" of green marketing innovatively, and provides a future research prospect.

Reviewer 3 Report (New Reviewer)

Dear authors,

I think that the paper is conceptually and content-wise not at the level of Sustainability journal.

However, I would not discourage the authors, but would also encourage them to create a broader picture, that is, a more detailed review of the literature, to present contemporary sources of literature, to refer to them and to determine in more detail the research area they are working on.

Author Response

Response to Reviewer 3 Comments

Point 1: I think that the paper is conceptually and content-wise not at the level of Sustainability journal. However, I would not discourage the authors, but would also encourage them to create a broader picture, that is, a more detailed review of the literature, to present contemporary sources of literature, to refer to them and to determine in more detail the research area they are working on.

Response 1:

Thank you very much for your suggestion. I have added the corresponding literature review section and further refined the data analysis and conclusion discussion. I look forward to more suggestions from you, and I will accept them with an open mind and adopt them seriously.

This is a reorganized abstract of the paper:

"Lucid waters and lush mountains are invaluable assets". Green marketing and sustainable development have become crucial topics in economic and social development. This study utilized the Cite Space software to visualize and analyze relevant literature published in the Web of Science core database from 2019 to 2023, drawing knowledge maps of collaborating authors, institutions, countries, and keywords to identify the research hotspots and development directions of green marketing. The research results show an increasing trend in the publication and citation frequency of green marketing-related literature from 2019 to 2023, with the highest citation frequency achieved in 2022. The relationship between Green management, Firm performance, Green innovation, Green strategy, Green capability, Sustainable development, Green business, Green market orientation, Green supply chain management, Green exploitation, Green responsibility, Ecosystem, Green commitment, Green perceptions, Green competitive advantage, Corporate social responsibility reflect important hotspots of green marketing research. From a microscopic perspective, the study sorts out concepts related to green marketing, discusses the virtuous ecological cycle of "individual-environment-economy-society" of green marketing innovatively, and provides a future research prospect.

Reviewer 4 Report (New Reviewer)

Specific comments 

This study shows the relationship sustainability-store equity, green innovation, entrepreneurial orientation, value creation, strategic capability, green practice, sustainable development, strategy, measuring corporate wrongdoing, management innovation, emerging market penetration management innovation, emerging market penetration, and green supply chain reflect the important hotspots of green marketing research. Authors focused on sorting out the concepts related to green marketing

In my view, the topic has the originality to be considered for publication as it covers the gap in the literature.

The authors provided a comprehensive literature review although some articles should be added.

The conclusion has written in a proper format.

Some missing references should be added (please see my comments).

Tables and figures are legible.

General comments

The manuscript entitled “Cite Space-based Bibliometric Analysis of Green Marketing” seems to be acceptable to be considered for publication in Sustainability, but it would be better if some issues be regarded in the revision.
1. Since this paper is not a literature review, in lieu of referring to a vast number of previous works, which have been surveyed by other researchers, mention the latest related works.

Oluwaseun Kolade. "Production network and emission control targets-theoretical approach." Peace Economics, Peace Science and Public Policy 29.1 (2023): 43-69.

Gao, Huichen, and Shijuan Wang. "The Intellectual Structure of Research on Rural-to-Urban Migrants: A Bibliometric Analysis." International Journal of Environmental Research and Public Health 19.15 (2022): 9729.

A two-stage stochastic planning model for locating product collection centers in green logistics networks." Cleaner Logistics and Supply Chain 6 (2023): 100091.

2. The transitions from topic to topic in the paper seem to be a little sudden. In other words, while reading about a topic, the text suddenly starts to mention something quite different. It is suggested to smooth these transitions from topic to topic where possible.
3. Please indicate the contributions in more detail, specifically in comparison with the latest research papers.
4. Above all, please polish the title to be in line with the contents of this manuscript.

Moderate editing of English language required

Author Response

Response to Reviewer 4 Comments

Point 1: This study shows the relationship sustainability-store equity, green innovation, entrepreneurial orientation, value creation, strategic capability, green practice, sustainable development, strategy, measuring corporate wrongdoing, management innovation, emerging market penetration management innovation, emerging market penetration, and green supply chain reflect the important hotspots of green marketing research. Authors focused on sorting out the concepts related to green marketing. In my view, the topic has the originality to be considered for publication as it covers the gap in the literature.

Response 1: Thank you very much for your suggestions. This study is original to the author and is dedicated to revealing the hotspots and trends of green marketing research in recent years, from posing questions, literature review, data analysis to conclusion discussion.

Point 2: The authors provided a comprehensive literature review although some articles should be added. Some missing references should be added (please see my comments).

Since this paper is not a literature review, in lieu of referring to a vast number of previous works, which have been surveyed by other researchers, mention the latest related works.

Response 2: Thank you very much for your suggestions. I noticed this problem. I have added literature related to this data analysis method in the data analysis section. For example:

[27] Chen C. CiteSpace II: Detecting and visualizing emerging trends and transient patterns in scientific literature[J]. Journal of the American Society for information Science and Technology, 2006, 57(3): 359-377.

[28] Liu S, Sun Y P, Gao X L, et al. Knowledge domain and emerging trends in Alzheimer’s disease: a scientometric review based on CiteSpace analysis[J]. Neural Regeneration Research, 2019, 14(9): 1643.

[29] Cui Y, Mou J, Liu Y. Knowledge mapping of social commerce research: A visual analysis using CiteSpace[J]. Electronic Commerce Research, 2018, 18: 837-868.

[30] Kleinberg J. Bursty and hierarchical structure in streams[C]//Proceedings of the eighth ACM SIGKDD international conference on Knowledge discovery and data mining. 2002: 91-101.

[31] Saleem F, Khattak A, Ur Rehman S, et al. Bibliometric analysis of green marketing research from 1977 to 2020[J]. Publications, 2021, 9(1): 1.

[32] Yang H, Shao X, Wu M. A review on ecosystem health research: A visualization based on CiteSpace[J]. Sustainability, 2019, 11(18): 4908.

[33] Oluwaseun Kolade. "Production network and emission control targets-theoretical approach." Peace Economics, Peace Science and Public Policy 29.1 (2023): 43-69.

[34] Gao, Huichen, and Shijuan Wang. "The Intellectual Structure of Research on Rural-to-Urban Migrants: A Bibliometric Analysis." International Journal of Environmental Research and Public Health 19.15 (2022): 9729.

Point 3: The conclusion has written in a proper format. Tables and figures are legible.

Response 3: Thank you very much for your comments, I have further optimized the concluding discussion section as well as the figures and tables in the paper.

Point 4: The manuscript entitled “Cite Space-based Bibliometric Analysis of Green Marketing” seems to be acceptable to be considered for publication in Sustainability, but it would be better if some issues be regarded in the revision.

Response 4: Thank you very much for your suggestion. I have added a literature review section and improved the results and textual description of the data analysis. Also, I improved the conclusion discussion and elaborated on the study limitations and future perspectives.

Point 5: The transitions from topic to topic in the paper seem to be a little sudden. In other words, while reading about a topic, the text suddenly starts to mention something quite different. It is suggested to smooth these transitions from topic to topic where possible.

Response 5: I noticed the problem. I enhanced the connection between paragraphs and refined the textual presentation.

Point 6: Please indicate the contributions in more detail, specifically in comparison with the latest research papers.

Response 6:  Thank you very much for your suggestion. I will separate the conclusion and the discussion. The conclusion section is mainly a summary overview of the results of the data analysis. The general discussion is divided into three parts, including theoretical combinations, research limitations and future prospects.

Point 7: Above all, please polish the title to be in line with the contents of this manuscript.

Response 7: Thank you very much for your suggestion. I reworked the English language for grammatical errors and retouching.

Round 2

Reviewer 1 Report (Previous Reviewer 1)

Overall, the revisions were done well, but it appears that the author did not address the writing of Chapter 5 and Chapter 6. These two chapters should include specific content and exclude certain content. Please refer to the format of previous research as a writing example.

In essence, Chapter 5 should be the discussion section, which should incorporate literature to highlight the theoretical and practical contributions of your research. It should critically elaborate on the research findings. Chapter 6 should be the conclusion section, divided into contributions, limitations, and future research.

Author Response

Point 1: Overall, the revisions were done well, but it appears that the author did not address the writing of Chapter 5 and Chapter 6. These two chapters should include specific content and exclude certain content. Please refer to the format of previous research as a writing example.

In essence, Chapter 5 should be the discussion section, which should incorporate literature to highlight the theoretical and practical contributions of your research. It should critically elaborate on the research findings. Chapter 6 should be the conclusion section, divided into contributions, limitations, and future research.

Response 1: Thank you very much for your suggestion. In the discussion section of Chapter 5 I discuss the findings in the context of the literature review and data analysis. In the conclusion section of Chapter 6, I refine it according to contribution, research limitations, and future research:

  1. General Discussion

5.1 Green Marketing Overall trends

Green marketing and sustainable development is an important topic for economic and social development. From the data of bibliometrics, there are 952 articles and 9723 citation data of related literature about green marketing in 2019-2023, and green marketing is paid attention by a large number of scholars. Similarly, the literature related to green marketing shows the development trend of increasing year by year, and the academic community still maintains a high attention to the research on green marketing. From the perspective of authors, institutions, and countries, the USA, CHINA, and ENGLAND, as countries with high influence in the field of green marketing research, the above three countries can be cited as important sources of green marketing literature. Indian Institute of Management, White Rose University of Consortium, University System of Ohio, University of System of Florida, Xi' a Jiaotong University, etc., as important authors with high citation rates in green marketing literature, the green marketing literature of the above institutions can be a good source for other green marketing researchers. The green marketing literature of these institutions can be a good inspiration and reflection for other green marketing researchers.

5.2 Green Marketing Keywords

Reviewing the analysis of the literature review, the study found that scholars mostly think about green marketing from enterprise (vs. consumer) perspectives [4].

The enterprise perspective of green marketing focuses on green marketing orientation [18, 19, 20, 21, 22], green marketing capability [23, 24, 25, 26, 27], and green marketing effectiveness [28, 29, 30, 31, 32], which mainly include corporate green marketing awareness and green marketing formulation and implementation. Further through bibliometrics analysis, it was found that keywords with high co-occurrence include corporate social responsibility, entrepreneurial orientation, environmental innovation, innovation governance, sustainable supply chain management, relational governance, supplier code, firm innovation, and sustainable development goal, which are in line with the enterprise perspective of current green marketing research. Furthermore, the high-frequency keywords of green marketing research include Green management, Firm performance, Green innovation, Green strategy, financial performance, Green capability, Sustainable development, Green Business, Green market orientation, and Green supply chain management. The high-frequency keywords for green marketing research include Green conceptual framework, Green responsibility, Green business performance, Green organizations, Green commitment, Green perceptions, Green market orientation, Green supply chain management, Green competitive advantage, and Corporate social responsibility. Therefore, high-frequency keywords (vs. high-centrality keywords) overlap with keywords such as Green market orientation, Corporate social responsibility, Green supply chain management, and Green business performance can be the key themes of green marketing focus. However, the existing studies of green marketing from an enterprise perspective have not deeply considered the possible external environmental pressures and the negative effects of market dynamics of green marketing, but the data results show the high impact intensity of Green uncertainty and slack resources keywords, which indicates that the uncertainty of resource scarcity and green development as the external environmental pressures of green marketing has an important role, which fills an important gap in green marketing research.

The consumer perspective on green marketing focuses on the possible green consumption decisions and purchase intentions of consumers [33, 34, 35, 36]. The overlapping keywords Green perceptions with high-frequency keywords (vs. high centrality keywords) also reflect the importance of green consumers in green marketing research, but the literature on green consumers is cited and co-occurrence less frequently.

  1. Conclusion

6.1 Theoretical implications

Based on the literature review and measurement, the study finds that the main subjects of green marketing research include enterprises and consumers. The study aims to innovatively sort out the virtuous cycle of "individual (consumer vs. company)-environment-economy-society" green marketing and sustainability, and to give a macroscopic understanding of the texts and data of literature studies.

From the perspective of consumers, green marketing is a new marketing model and marketing is practice-oriented. The practice results of green marketing depend on the two-level attitudes accepted by consumers, which are influenced by consumer cognition, emotion, and behavior. Green moral motivation drives consumers' environmental moral identity and green personal norms [44], and consumers identify, understand, and actively generate pro-environmental behaviors, consciously constraining their green responsibility awareness and behavioral norms. For green products, consumers' perception quality and perception value directly affect their behavioral performance. Green perception quality refers to consumers' overall perception and judgment of the quality or other superiority of green products and services provided by companies [45, 46], while green perception value refers to the gains and losses perceived by consumers when making purchasing decisions for green products [47]. Both perception quality and perception value significantly affect consumers' green consumption preferences and purchase intentions, thereby achieving a virtuous ecological cycle of "consumer-environment-economy-society".

From the perspective of enterprises, the green marketing performance and sustainable development configuration of enterprises are significantly affected by green marketing capability (GMC), which is the ability of enterprises to identify, acquire, convert, and integrate existing resources to create products and services that meet environmental requirements of enterprises, with two key elements of green market perception and green marketing implementation [23, 25]. Green market perception is the overall planning and strategic goal setting of enterprises for the green market, and timely learning based on dynamic changes in the market such as consumers and competitors [48]. From an overall strategic perspective, enterprises fully consider corporate social responsibility, sustainable strategic goals, and sustainable development. Green marketing implementation achieves the established goals of enterprises through market marketing strategies, including green product design and development, green pricing considering economic and environmental costs, green distribution channels for sustainable supply chain management, green promotion methods that convey information such as green and environmental protection of enterprise products to the outside world, as well as integrating cross-functional sustainable innovation and development capabilities within the enterprises [49]. Based on macro and micro measures such as environmental recognition, strategic formulation, and marketing implementation, enterprises achieve a virtuous ecological cycle of "enterprise-environment-economy-society".

6.2. Research limitations

The study also has certain research limitations. On the one hand, the research objects mainly include SCI and SSCI source journals in the Web of Science core database in the past five years, with reliable data sources and convincing data analysis results. However, the research data mainly analyzed authors, institutions, countries, and keywords. Due to the time limit, the study failed to sort out the timeline of green marketing research and did not comb the development context of green marketing research. On the other hand, the analysis of existing literature using Cite Space software belongs to the objective analysis category and is not influenced by personal subjective experience. However, there may still be data errors in the literature research.

6.3 Future perspectives

The study also provides new thinking for the future of green marketing. By analyzing the sudden increase or decrease in the number of keyword citations in a specific period, the study found that Eco-innovation, Sustainable innovation, Green scale development, Green ownership, Green growth, Business relationships, Manufacturing firms, and Green markets have a longer time duration of impact. Therefore, in addition to a green marketing orientation, green marketing capability, green marketing effectiveness, and green consumers, scholars can think about the impact of Eco-innovation and sustainable innovation on individuals (enterprises vs. consumers), the economy, and society in future studies.

Reviewer 3 Report (New Reviewer)

The paper has been significantly improved, however, in order to reach the final version, the following needs to be done:

The Abstract should have the following logic:

Purpose: Design / methodology / approach; Findings; Practical implications; Originality

The purpose of the study is also unclear. I recommend redoing the  introduction, with the following in mind: 

1- Make a frame for the reader

2- Issues of the topic under analysis

3- Evidence of the GAP of the literature based on the literature

4- Purpose of the study

5- Originality of the study

6- What are the expected results (to captivate the reader)

7- The last paragraph should briefly describe what the reader can read in the following sections.

In discussing the results, the authors do not confront them with the literature.

The conclusion must have:

Remember the objective of the study

Main findings

Theoretical and practical implications

Originality of the study

Study limitations

Future lines of research

The study's bibliography is weak. There is a lack of studies with relevance to the topic and quality.

Author Response

Point 1: The Abstract should have the following logic:

Purpose: Design/methodology / approach; Findings; Practical implications; Originality

Response 1: Thank you very much for your suggestion. I have regrouped the abstract according to a certain logic:

"Lucid waters and lush mountains are invaluable assets". Green marketing and sustainable development have become crucial topics in economic and social development. During 2019-2023, there are nearly 952 publications on green marketing-related topics in the Web of Science core database, and a large number of scholars researched green marketing, but there is still a lack of comprehensive and systematic studies on the current status of green marketing research and hot trends. The main purpose of this study is to summarize and sort out the current status of green marketing research by reviewing the literature related to green marketing (enterprises vs. consumers). Meanwhile, using Cite Space software, the core literature of green marketing in the past five years was screened and data visualized and analyzed, and the knowledge map of cooperative authors, institutions, countries, and keywords was drawn, in an attempt to discover the research hotspots and major development trends of green marketing. The results show that the publications and citation frequency of green marketing-related literature show an increasing trend from 2019 to 2023, and the highest citation frequency is reached in 2022. Green marketing literature appears more frequently in the USA, CHINA, and ENGLAND, and has a higher influence. Green management, Firm performance, Green innovation, Green strategy, Green capability, Sustainable development, Green Business, Green market orientation, Green supply chain management, Green exploitation, Green responsibility, Ecosystem, Green commitment, Green perceptions, Green competitive advantage, and Corporate social responsibility reflect the hot topics and important themes of green marketing research. This study innovatively combines green marketing literature review and bibliometric analysis, comprehensively and systematically discusses the current status of green marketing research and hot trends based on both textual and data perspectives, and puts forward the "individual-environment-economy-society" virtuous ecological cycle of green marketing from a macro perspective.

Point 2: The purpose of the study is also unclear. I recommend redoing the introduction, with the following in mind: 

Response 2: Thank you very much for your suggestion. I have reworked the introduction to follow the logic:

  1. Introduction

"Lucid waters and lush mountains are invaluable assets" is an important assertion made by Chinese President Xi Jinping that aligns with the significant issue of green marketing and sustainable development. In 2022, the resumed Fifth United Nations Environment Assembly (UNEA5.2) called on the international community to unite for a peaceful and healthy environment, focusing on global economic recovery and sustainable development and emphasizing the importance of strengthening nature's actions to achieve Sustainable Development Goals, which are key to personal, economic, environmental, and social sustainability. Focusing on the marketing perspective, green marketing and sustainability have received a great deal of attention from scholars and are important hot trends in green marketing research in the new era.

Green marketing includes ecological marketing, environmental marketing, marketing of resource-saving and environmentally friendly products [1], and marketing that meets the needs of consumers and enterprises in the present while preserving or enhancing inter-generational legacy [2]. Green marketing integrates personal, economic, and social considerations and is an important tool and marketing approach to promote resource conservation, environmental protection, and ecological development. From the enterprise perspective, green marketing focuses on corporate social responsibility [3], sustainability marketing [4], socially responsible marketing [5], green marketing orientation [6], green marketing [7], corporate green product innovation [8], green marketing strategies [9], and so on. The research topics are very rich, but there is a lack of influence of other social subjects such as government, consumers, the public, and corporate employees on green marketing. There is also no research to think deeply about the possible negative effects of external environmental pressure and market dynamics of green marketing. From the perspective of consumers, green marketing is mainly manifested in consumers' awareness and behavior of practicing green consumption, and some scholars point out that consumers' pro-environmental behavior [10, 11] and consumers' innovativeness [12, 13] influence consumers' willingness to green consumption. However, few studies have explored the deviation of green consumption behavior from consumers' actual purchase, as well as consumers' willingness to repeat green consumption, and there is a lack of research on consumers' repeated sustainable consumption of green products.  

Based on the importance of green marketing research and the current status of existing research, scholars have gradually developed quantitative research methods for literature reviews by shifting from qualitative analysis of literature reviews to econometric analysis of literature. Initially, most scholars conducted the review analysis of green marketing through literature combing [14]. With the research development of the bibliometric analysis method, the quality and quantity of literature are visualized by analyzing literature authors, publishing institutions, countries, and citation frequency [15]. Bibliometric analysis intuitively and concisely reflect the existing research trends and hot topics, but few scholars use data visualization analysis for green marketing research [16], and few studies consider green marketing from a macro perspective of sustainability.

Therefore, the study aims to use the data visualization software Cite Space, using the SCI and SSCI source journals of the Web of Science core database as the main data source, and "Green marketing" or " Environmental Marketing" or "Lower- Carbon Marketing" or "Ecological Marketing " or "Sustainable Development" as a broad keyword search to conduct a comprehensive and systematic bibliometric study of green marketing. Through the mapping of co-authors, institutions, countries, and keywords, this study innovatively combines green marketing literature review and literature data analysis to discuss the current status of green marketing research and hot trends comprehensively and systematically based on both textual and data perspectives, providing practical references for subsequent scholars' research selection and green marketing theory development. At the same time, the study will also specifically explore the following issues:

(1) What is the current status of research on green marketing research?

(2) What is the overall trend of green marketing research from 2019-2023?

(3) Which authors, publishers, and countries have a higher influence in the field of green marketing?

(4) What are the hot keywords in green marketing?

(5) What are the future trends in green marketing research?

Point 3: In discussing the results, the authors do not confront them with the literature.

The conclusion must have:

Remember the objective of the study

Main findings

Theoretical and practical implications

The originality of the study

Study limitations

Future lines of research

The study's bibliography is weak. There is a lack of studies with relevance to the topic and quality.

 Response 3: Thank you very much for your suggestion. In the discussion section of Chapter 5, I discuss the findings in the context of the literature review and data analysis. In the conclusion section of Chapter 6, I refine it according to the contribution, research limitations, and future research.

Similarly, because the main research is focused on green marketing, there are more references to literature about green marketing. In conjunction with Cite Space's bibliometric analysis, I have censored some of the literature and added literature relevant to bibliometrics.

Round 3

Reviewer 1 Report (Previous Reviewer 1)

The author responded to my comments.

I hope that in the future, you can continue to strive and explore more necessary topics using more in-depth research and analysis methods, making greater contributions to society.

Congratulations!

This manuscript is a resubmission of an earlier submission. The following is a list of the peer review reports and author responses from that submission.

Round 1

Reviewer 1 Report

The title of this manuscript indicates that it is a review paper, but the author's submission type is listed as an article. Please ensure consistency in the description of your research type.

For a review paper, this paper lacks an adequate number of references, and many of them are not from the past five years, which is inadequate. I believe the author's handling of the references is quite weak.

Please ensure to provide the full names before using any abbreviations.

In Table 1, the author appears to list the basis for literature search, but there is no further analysis of the found 700+ articles. The author only attempts to summarize the research directions of these articles and presents Figure 1. This seems to be the extent of the contribution in this paper.

It is unclear what "Centrality" represents in Table 2.

The author does not explain on what basis the 11 articles in Table 2 were selected from the 700+ articles, nor is there any further analysis of these 11 articles. Only the titles are listed, and I do not understand the purpose of doing so.

Figures 1, 2, 3, and 4 appear well-designed, but unfortunately, they provide extremely limited information. These figures only demonstrate what kind of topics can be found based on the search keywords chosen by the author, and that's it. Especially the presentation of Figures 2 and 4 is visually impressive but lacks practical significance. The author also does not provide further explanations or deeper analysis in the text.

In the fourth chapter, the author seems to discuss the literature on several well-known quantitative research models. I fail to understand the purpose of doing so. These quantitative models are widely known, and the author merely provides a simple explanation of their definitions, which does not add much value to your review paper on sustainable marketing.

Lastly, I would like to point out that a review paper is generally better suited for researchers with experience in the field. It is challenging for students to accurately grasp the key points of the industry and write a high-quality review with a mature writing style. I suggest that the author consider starting with a valuable article and gradually learn the art of academic writing.

Reviewer 2 Report

In the research, keywords related to green marketing and sustainability between 2018-2023 were scanned and a bibliometric study was conducted.

The findings related to the research are as follows;

1. The scope of the research is quite limited. And it is not original. There are more extensive studies on this subject.

2. In the research, there are only the findings obtained by keyword scanning, and there is no discussion that provides added value such as any comments or correlations.

3. The work done within the scope of this research is already obtained automatically in the relevant database.

4. The research has not been found original. It can be re-evaluated if it is developed by the authors.

Good luck to the author/s.

  In the research, keywords related to green marketing and sustainability between 2018-2023 were scanned and a bibliometric study was conducted. The findings related to the research are as follows; 1. The scope of the research is quite limited. And it is not original. There are more extensive studies on this subject. 2. In the research, there are only the findings obtained by keyword scanning, and there is no discussion that provides added value such as any comments or correlations. 3. The work done within the scope of this research is already obtained automatically in the relevant database. 4. The research has not been found original. Good luck to the author/s.

Minor problem.

Reviewer 3 Report

In general, the article is written chaotically. The Authors did not specify the purpose of the article (research) or research questions. The significance of the research is not defined. There are a lot of generalities in the article. The research method is not described in detail, which is not surprising, considering that the article does not have a specific purpose. The article is too short, the literature review is not in-depth.  The description of green marketing appears in several different places in the article.  The article is not written in accordance with the journal template. "Lucid waters and lush mountains are invaluable assets" this sentence is a quote from Xi Jinping and it should be noted in the text, firstly because it is a quote, secondly, outside of China, not every reader needs to know this assertion.

I think that the Authors have an interesting research idea and thoughts, but they need to organize and refine them.